



# Investigating the interaction of waves and river discharge during compound flooding at Breede Estuary, South Africa

Sunna Kupfer[1], Sara Santamaria-Aguilar[1], Lara van Niekerk[2,3], Melanie Lück-Vogel[2,4], Athanasios T. Vafeidis[1]

[1]Coastal Risks and Sea-Level Rise Research Group, Department of Geography, Christian-Albrechts University, Kiel, Germany
[2]Coastal Systems Research Group, Council for Scientific and Industrial Research CSIR, Stellenbosch, South Africa
[3]Institute for Coastal and Marine Research, Nelson Mandela University, PO Box 77000, Port Elizabeth, 6031, South Africa
[4]Department for Geography and Environmental Studies, Stellenbosch University, Stellenbosch, South Africa.

*Correspondence to*: Sunna Kupfer (kupfer@geographie.uni-kiel.de)

**Abstract.** Recent studies have drawn special attention to the significant dependencies between flood drivers and the occurrence of compound flood events in coastal areas. This study investigates compound flooding from tides, river discharge (Q) and specifically waves using a hydrodynamic model at Breede Estuary, South Africa. We quantify vertical and horizontal differences in flood characteristics caused by driver interaction, and assess the contribution of waves. Therefore, we compare flood characteristics resulting from compound flood scenarios to those in which single drivers are omitted. We find that flood characteristics are more sensitive to Q than to waves, particularly when the latter only coincide with high spring tides. When interacting with Q, however, the contribution of waves is high, causing 10-12 % larger flood extents and 45-85 cm higher water depths, as waves caused backwater effects and raised water levels inside the lower reaches of the estuary. With higher wave intensity, the first flooding began up to 12 hours earlier. Our findings provide insights on compound flooding in terms of flood magnitude and timing at a South African estuary and demonstrate the need to account for the effects of compound events, including waves, in future flood impact assessments of open South African estuaries.

## Introduction

Floods, regardless of fluvial or oceanic origin, are among the world's most devastating coastal hazards, causing numerous deaths and large economic losses on an annual basis (Kirezci et al., 2020). Despite improved flood protection, forecasting, and warnings, flooding remains a growing threat, due to the continued global coastal urbanisation which result in rapid population growth, economic development and land use change (Brown et al., 2018; Hallegatte et al., 2013; Hanson et al., 2011). Moreover, the accelerating rate of sea-level rise (SLR) may cause historically rare floods to become common by the end of the century (Vitousek et al. 2017). In coastal areas, the interactions of oceanographic, hydrological, and meteorological phenomena can lead to extensive flooding. Particularly in estuaries, such floods can result from combined spring tides and extreme wave or storm surge conditions occurring simultaneously with high river discharge (Kumbier et al., 2018; Olbert et al., 2017; Ward et al., 2018). These events are commonly referred to as compound flood events. Definitions of compound events have evolved





in recent years (Leonard et al., 2014; Zscheischler et al., 2018; Couasnon et al., 2020; IPCC, 2014) and these events are described as incidents that result from the combination of physical drivers, leading to stronger impacts than from drivers, occurring individually. Thus, neither of the drivers needs to be extreme in order to cause severe impacts, as drivers that occur simultaneously or successively can result in extreme events, which contribute to societal or environmental risk (Leonard et al.,
2014; Seneviratne et al., 2012; Zscheischler et al., 2018).

Recent global and regional joint-probability analysis of river discharge, storm surge and waves (Couasnon et al., 2020; Ward et al., 2017; Hendry et al., 2019; Wahl et al., 2015); as well as local-scale case studies distributed around the globe (Mazas and Hamm, 2017; Bevacqua et al., 2019; Klerk et al., 2015; Rueda et al., 2016); have drawn special attention to statistical dependencies between flood drivers and higher occurrence probabilities of compound events with climate change.
With climate-change-induced sea-level rise (Nerem et al., 2018),  potential changes in storminess (Church et al., 2013), more extreme precipitation (Myhre et al., 2019) and higher river discharge (van Vliet et al., 2013), the risk of compound flooding is likely to increase, and flood extent, magnitude and duration can be locally exacerbated (Couasnon et al., 2020).

Despite such studies focussing on dependencies between flood drivers, few published research on compound flood assessments exist, with most exploring the differences in flooding caused by the interaction of fluvial drivers with storm surge and tides
(e.g. Olbert et al., 2017; Kumbier et al., 2018; Chen and Liu, 2014), pluvial drivers with surge (e.g. Bilskie and Hagen, 2018; Bilskie et al., 2020) and tides (e.g. Shen et al., 2019). These studies successfully address the driver interaction in hydrodynamic models and highlight the improved understanding of flood dynamics, when considering the interaction of flood drivers (Olbert et al., 2017; Lee et al., 2020; Shen et al., 2019; Seenath et al., 2016). When coinciding with high river discharge, the contribution of waves to flooding are seldom addressed (e.g. Lee et al., 2020), even though waves play a substantial role in
terms of flooding in many of the discussed areas (Kumbier et al., 2018; Bilskie and Hagen, 2018), while the influence on the timing of the flood has not been analysed in detail.

The South African coastal area is particularly vulnerable to wave-induced flooding as cold fronts, cut-off lows, and cyclones cause large destructive swells (Guastella and Rossouw, 2012), which constitute the most important components of coastal
floods in South Africa (Theron et al., 2010). These low-pressure systems cause additional heavy rainfalls, leading to immense fluvial flash floods (Pyle and Jacobs, 2016; Molekwa, 2013).

The South African coastline comprises 291 estuaries, with the majority of rapidly developing coastal towns situated around estuaries (Hughes and Brundrit, 1995; van Niekerk et al., 2020). Since estuaries are potentially prone to flooding from fluvial
and coastal high water-levels, urban development in and around estuaries may be affected from compound flooding (Pyle and Jacobs, 2016). For this reason, in 2019-2020, the South African Department for Forestry, Fisheries and Environment conducted the National Coastal Climate Change Assessment, which addressed coastal and estuarine flooding (DEFF, 2020); however, this study did not account for compound flooding.





Currently, no published regional to local compound flood probability analyses exist for South Africa and global statistical
dependency analysis accounting for storm surge and river discharge only show small correlations between drivers (Couasnon
et al., 2020). This may be due to the fact that the surge contribution compared to other flood drivers, such as tides and waves
is relatively small in most South African estuaries (Theron et al., 2010; Theron and Rossouw, 2008). However, as extreme
precipitation and waves are generated from the same atmospheric forcing in South Africa, a dependency between both drivers
is likely. Yet compound flood effects have not been addressed for South Africa in the published literature. Flood impact
assessments in general are rare, and those documented mostly assess the flood drivers individually (Fitchett et al., 2016;
Mather and Stretch, 2012; Theron et al., 2010).

The main objective of this study is to analyse local scale compound flooding at Breede Estuary, a South African permanently
open estuary. Thereby we specifically account for the contribution of waves when coinciding with high river discharge. In this
context we assess the effects of compound flooding from river discharge, tides, and waves in terms of magnitude and timing
on the lower estuary, by using the hydrodynamic model Delft3D. We analyse the interaction of all drivers and estimate the
sensitivity of the flood characteristics (extent, depth, and timing) to various driver combinations and intensities. We chose
Breede Estuary as it has a large catchment, a notable tidal exchange and data could be obtained. Finally, the lower estuary has
shown to be prone to flooding from coastal and fluvial drivers (see Basson et al., 2017) and since we focus on the contribution
of waves during compound flooding, our study site is constrained to the lower estuary.
The paper is structured as follows. We describe the characteristics of Breede Estuary in section 2. We explain the hydrodynamic
model set-up, data used and compound event scenarios in section 3. In section 4 we present flood characteristics, resulting
from the compound event scenarios, which we discuss in section 5.

## 2. Study Area

Of South Africa's 291 estuaries, Breede Estuary is one of the largest permanently open estuaries (van Niekerk et al., 2020).
Breede River has the fourth largest annual runoff in South Africa (Taljaard, 2003). It flows along 322 km from the south-west
of the country, in south-easterly direction towards the South African south coast and enters the Indian Ocean at the town
Witsand in Sebastian Bay (Fig. 1). The estuary extends about 50 km upstream, where the tidal influence ceases (Lamberth et
al., 2008).


Breede Estuary is sparsely populated by small settlements of up to 1000 inhabitants (e.g., Witsand, Fig. 1) situated on the
northern and southern banks. The estuary provides tourism services with several holiday resorts located along the banks.
Numerous farm properties spread along the banks further upstream, and most of the land in the immediate surroundings is
privately owned agricultural land (SSI, 2016).




Breede Estuary is open towards the south-east, where it enters the sea against a wave-cut terrace (Carter, 1983). Its mouth is characterised by an open channel, which is located at the southern end of an extensive sand barrier, formed by wave action (Schumann, 2013). Over the first 28 km, the depth of the estuary channel ranges from 3 to 6 m (SSI, 2016). At the lower estuary, the channel meanders along large and shallow sand banks, which have formed along the southern shore (Fig. 1).

During the low-flow summer months, the estuary is marine dominated, meaning the estuary receives high seawater input (SSI, 2016). Due to the relatively strong tidal inflow during summer (Taljaard, 2003), and the sand barrier, restricting the estuarine inlet, the estuary can be classified as tide and wave dominated (Cooper, (2001).

The main tidal signal is semi-diurnal (M2), with additional diurnal oscillations (Schumann, 2013). During spring tidal periods,
the tidal range can reach up to 2 m, as measured at the tide gauge of Witsand, situated at the northern shore of Breede Estuary (Fig. 1). The southern coastline is wave dominated and experiences the highest wave conditions along the entire South African coast (Theron et al., 2010). Thus, waves cause the largest relevant contribution to extreme water levels (WLs) in South Africa (Melet et al., 2018). Such wave conditions are generated mainly by two synoptic weather systems, namely cold front systems and cut-off lows (Mather and Stretch, 2012). These are responsible for long-period to local swell conditions, with waves
approaching the south coast from south-westerly directions. Generally, annual mean significant wave heights ($H_s$) range from 2.4 - 2.7 m (Basson et al., 2017). During extreme storm events significant wave heights can reach more than 10 m and peak periods range from 5 s to 20 s. The estuary mouth is relatively sheltered from south-westerly waves since it is protected by a southern headland of the bay (Fig. 1). Waves from the south-eastern sector occur as well, however these are generated by tropical cyclones, making landfall at the Mozambican and the South African east coast (DEA&DP, 2012). The dominating
wind direction is from the westerly and easterly sector, whereby easterly winds generate local wind waves, penetrating into the estuary, as its opening faces east (Vonkeman et al., 2019). One example of coastal flooding occurring in the area, was an extreme storm in August 2008. Waves of 10.7 m were measured, and since the storm lasted longer than 12 hours, the extreme waves additionally co-occurred with high tide levels, one day after a spring tide. Consequently, a large area of the South African south coast was affected, resulting in severe damage to coastal infrastructure (Guastella and Rossouw, 2012).
During winter, the estuary is highly responsive to freshwater inflows (Taljaard, 2003). The catchment receives 80% of the annual rainfall during winter months, causing peak flows and floods usually during that season. Breede Estuary has experienced extreme fluvial flooding, with major events occurring in 1906, 2003 and 2008. In November 2008, intense rainfall far upstream, caused by a cut-off low, resulted in extreme river runoff (Holloway et al., 2010). Extreme river discharge caused WLs up to 10 m in the upper 20 km of the estuary while levels of 50 cm were measured at the estuary entrance (Basson et al., 2017). A
similar cut-off low event occurred in May 2021 but was less extreme, with estimated elevated WLs being 1-2 m in the upper reaches.




**Figure 1.** Location of the study area and aerial photographs showing the Breede River and the Breede Estuary.

## 3. Methods

### 3.1 Hydrodynamic model and data description

We used the fully integrated open source modelling suite Delft3D (Lesser et al., 2004) which has been extensively used in coastal applications (Lyddon et al., 2018; Bastidas et al., 2016; Kumbier et al., 2018) for simulating flood extents and flood depths from waves, tides and river discharge, hereafter referred to as Q. We used the hydrodynamic numerical module Delft3D-FLOW, coupled with the module Delft3D-WAVE, which is based on the SWAN (Simulate WAves Nearshore) model.

Setting up a hydrodynamic model requires numerous input datasets. The characteristics of the datasets used in this study are shown in Table 1. A detailed description of the pre-processing of the datasets used as Delft3D input files and the model set-up is provided in Appendix A.



**Table 1:** Datasets and characteristics applied to set up Delft3D.

| Data Set | Source | Horizontal Resolution | Temporal Resolution | Time period | Reference System |
|---|---|---|---|---|---|
| Bathymetry | Basson et al. (2017) | 5 m | - | - | MSL |
| Elevation SUDEM | van Niekerk (2016) | 5 m | - | - | MSL |
| Land Cover/ Bottom Roughness | DEA (2015)/* | 30 m | - | - | - |
| Tides FES2014 | AVISO (n.d.) | 1/16° | 1 hour | 1980-2014 | MSL |
| Q | H7H006 (DWS) | - | 1 hour | 1966-2019 | Local MSL |
| Waves | Basson et al. (2017) | - | constant | - | - |
| Observations | H14T007 (DWS) | - | 1 hour | 2002-2019 | Local MSL |

*Kaiser et al. (2011), Jung et al. (2011), Wamsley et al. (2009), Mourato et al. (2017), Chow (1959)


We performed simulations using tides and Q as input data in Delft3D-FLOW on a 5 x 5 m rectangular grid in a depth-averaged mode. As we focus on the additional contribution of waves during compound flooding, the model domain is restricted to the lower estuary (Fig. 2). Topographic input data were merged with bathymetric data, which were manually digitised, based on a bathymetry of an existing study report on flood lines at the Breede Estuary (Basson et al., 2017). We specified spatially

varying manning bottom roughness via literature review from gridded land cover data (Table 1). We obtained 17 years of hourly measured WL observations serving for the model calibration and validation from the tide gauge station H14T007 (DWS, n.d.a), located in the small harbour of the town Witsand (Fig. 2).

We forced the model at two open boundaries. The ocean boundary (Fig. 2) is located at the westernmost edge of the model domain and perpendicular to the main flow direction. Depending on the scenario, we forced this open boundary with tides and

waves. We used historical tidal input data (Table 1), which were obtained from the global tidal FES2014 model (AVISO, n.d; Carrère et al., 2015). The data were extracted at a point closest to, but still located 24 km offshore from the westernmost edge of the model domain (Fig. 2). The second boundary (upstream boundary, Fig. 2) is situated at the upstream border of the model domain, perpendicular to the river flow, and was forced by hourly measured Q from the station in Swellendam (Table 1), which was the closest to the upstream boundary (54 km). For the Delft3D-WAVE setup, we increased the grid cell size and

the horizontal resolution of the input bathymetry to 10 m for computational reasons. Since nearshore wave time series could not be obtained, a constant sea state (constant $H_s$ and constant $T_p$) serves as wave boundary conditions (ocean boundary, Fig. 2) which we obtained from two extreme value analysis (EVA), performed by Basson et al. (2017).


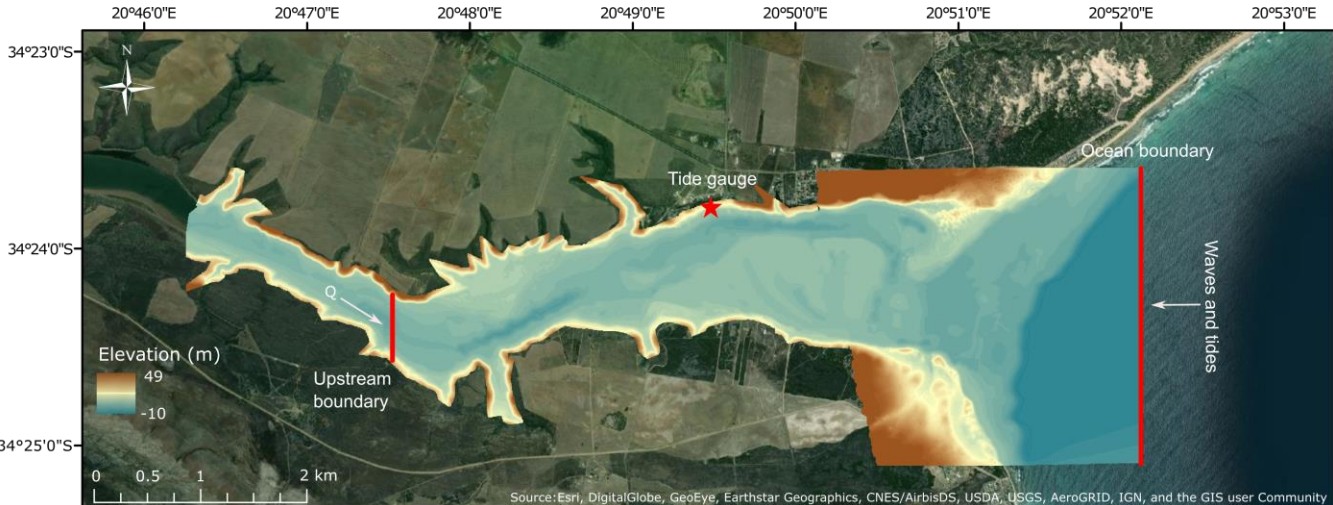

**Figure 2.** Model domain, including the merged bathymetry and elevation raster, the location of the Witsand tide gauge and the two open boundaries.

## 3.2 Model calibration and validation

To evaluate the performance of the model, we calculated the goodness-of-fit parameters $R^2$ (coefficient of determination), the Pearson correlation, $r$, and the root mean square error (RMSE) between the model output and observed WL time series (see Skinner et al., 2015).

During model calibration, we adjusted the bottom roughness, and horizontal eddy viscosity (see Appendix A). We used the best fitting physical parameters to set up the model for model validation and the scenario runs. Waves were excluded during model calibration and validation, since no measured nearshore wave time series could be obtained.

For the validation, we performed three simulations covering the full tidal range, and compared the model output to the corresponding observed WLs (Matte et al., 2017; Muis et al., 2017). To account for the full tidal range, these simulations include a spring-, average-, and a neap tide event. For these simulations we selected events, where Q was constantly low in order to focus on model performance when the model is driven only from the ocean boundary and where continuous observations exist. The largest spring tide event occurred from 27 September - 01 October 2007, the neap tide event from 18 - 23 September 2007 and the average tide event at 14 -19 July 2007. To test the performance of the model when driven by both the oceanic and the upstream river boundary, we selected the largest continuously recorded high Q event occurring within the period of observed WLs at the tide gauge in Witsand (22 - 25 November 2007). According to the tide gauge data, this high Q event (1262.78 m³s⁻¹) occurred simultaneously with a relatively large tidal range of up to 1.6 m.

For this validation event (21 - 27 November 2007) the time lag of Q reaching the upstream open boundary from the measuring station must be considered. Thus, we estimated the difference between the timing of the peak from the upstream flow gauge and from the non-tidal residual (NTR, see Appendix C) of the tide gauge, whereby we considered the maximum WL as the peak, caused by Q, since the tidal phase at this stage was at low tide level. We estimated a time lag of 8 hours, with the peak





at the tide gauge occurring later (Fig. C3). We accounted for this time lag in the Q boundary conditions for the validation run, to enable the comparison of model output and tide gauge data.

### 3.3 Event selection and scenario development

To assess compound flooding in terms of magnitude and timing, we developed four scenarios, accounting for tides, waves, and Q.

Storm surge was not considered, as no nearshore WL time series could be obtained, and offshore input data would even increase model uncertainties. Additionally, analysis of tide gauge data along the South African coastline has shown that at the South African south coast storm surge has a small contribution, relative to the other considered flood drivers, even when considering extreme surges such as a 100-year event (Theron and Rossouw, 2008; Theron et al., 2014). Moreover, Melet et al. (2018) showed that the wave contribution to extreme WLs in South Africa is substantially larger, compared to the surge contribution.

To explore this further, we additionally estimated the NTR of the tide gauge data of Witsand, which showed that the mean amplitude of the NTR of 10 cm is small compared to the tidal range of 2 m (Fig. A1). The contribution of wave set-up and river discharge is still included in the NTR, and large peaks could be identified as caused by river discharge (see Fig. C2 and more information on the analysis in Appendix C).

To investigate the effects of river discharge and waves on the flood characteristics during compound flooding, we developed

the following scenarios (Table 2):

**Table 2.** Scenario descriptions

| Scenario | River discharge | Tide | Waves |
|---|---|---|---|
| $S_{TQ}$ | 100-year (long) | Spring | - |
| $S_{TW}$ | Constant-low | Spring | 100-year (ESE direction) |
| $S_{TWQ}$ | 100-year (long) | Spring | 100-year (ESE direction) |
| $S_{TQWextr}$ | 100-year (long) | Spring | 100-year (all directions) |

The scenarios were named according to their driving mechanisms. Thereby $_T$ stands for tides, $_W$ for waves and $_Q$ for river

discharge. The selected extremes were extracted either via peak-over-threshold (POT) analysis, or by finding the maxima in the time series. The maximum Q event within the hourly river discharge time series applied for this study has a peak value of 1357 $m^3s^{-1}$ and occurred in November 2008. According to Basson et al. (2017) this value was corrected to 1546 $m^3s^{-1}$, corresponding to a return period of 15 years. Based on this value, a peak Q of 3295 $m^3s^{-1}$ corresponds to a 100-year event which we selected here as extreme Q. For low Q (used in scenario $S_{TW}$) an event where river discharge does not exceed 3 $m^3s^{-1}$

$^1$ was selected from the time series. For the spring tide event, we selected the maximum tidal flood peak of 1.3 m from the FES2014 tidal input data, which occurred in March 2007.

For the wave conditions, we chose two 100-year wave events from two different extreme value analysis (EVA) of Basson et al. (2017). According to their EVA, a 100-year wave event coming from east-south-easterly (ESE) directions (110°), the





direction from which waves directly penetrate the estuary, has a significant wave height of 6.2 m and a peak period of 12 s.
To consider an even higher wave event for a final worst-case scenario, H$_s$ was increased to 9.3 m and T$_p$ to 19.95 s,
corresponding to the significant wave height and peak period of a 100-year wave event, when considering all wave directions
in the EVA. The ESE wave direction was maintained for all scenarios that include waves. For the sea states driving the model,
it must be pointed out that Basson et al. (2017) performed EVAs on offshore wave data. As the location of the open boundary
for this study is located nearshore, the considered wave scenarios may be more extreme than the sea state would be at the open
boundary, as wave refraction and diffraction were not accounted for.

To compare the results of the scenarios WLs, flood extents and flood depths were extracted at the time of the maximum flood.

## 4. Results

### 4.1 Model validation

For all validation runs the model set-up was able to reproduce the timing of flood and ebb tide (Fig. 3). Variations occurred
however in the WL magnitude, especially during high tide (Fig. 3, upper left panel), where simulated WLs were 25 cm higher
than the observed for average tidal conditions. During low tide events in the spring tide simulation, modelled WLs were up to
60 cm lower (Fig. 3, lower left panel). The neap tide event on the other hand, was simulated with a RMSE of only 10 cm (Fig.
3, upper right). The goodness-of-fit estimates also showed agreement of observed and modelled WLs, for all tidal events,
excluding river discharge (Table 3).


**Table 3.** Goodness of fit estimates of model validation runs compared to observations for tide-only conditions and tides including high river discharge.

| Goodness-of-fit | Spring-, neap-, average tide | Spring-, neap-, average tide + high Q |
|---|---|---|
| RMSE | 0.21 m | 0.62 m |
| R² | 0.8 | 0.56 |
| r | 0.9 | 0.57 |

Moreover, for the simulation that included high Q (Fig. 3, lower right) the compared maximum WL peak did not show any
difference. After the maximum event peak, however, the model overestimated flood peaks by about up to 30 cm. WLs during
low tide before the peak of the event were strongly underestimated (~70 cm) by the model. The goodness-of-fit also
substantially decreased (Table 3).

As flooding is usually caused by peak WLs and simulated peaks showed an RMSE of 0.17 cm compared to observations for
all validation runs, we considered the model performance as fit-for the purpose of the analysis of flood events.
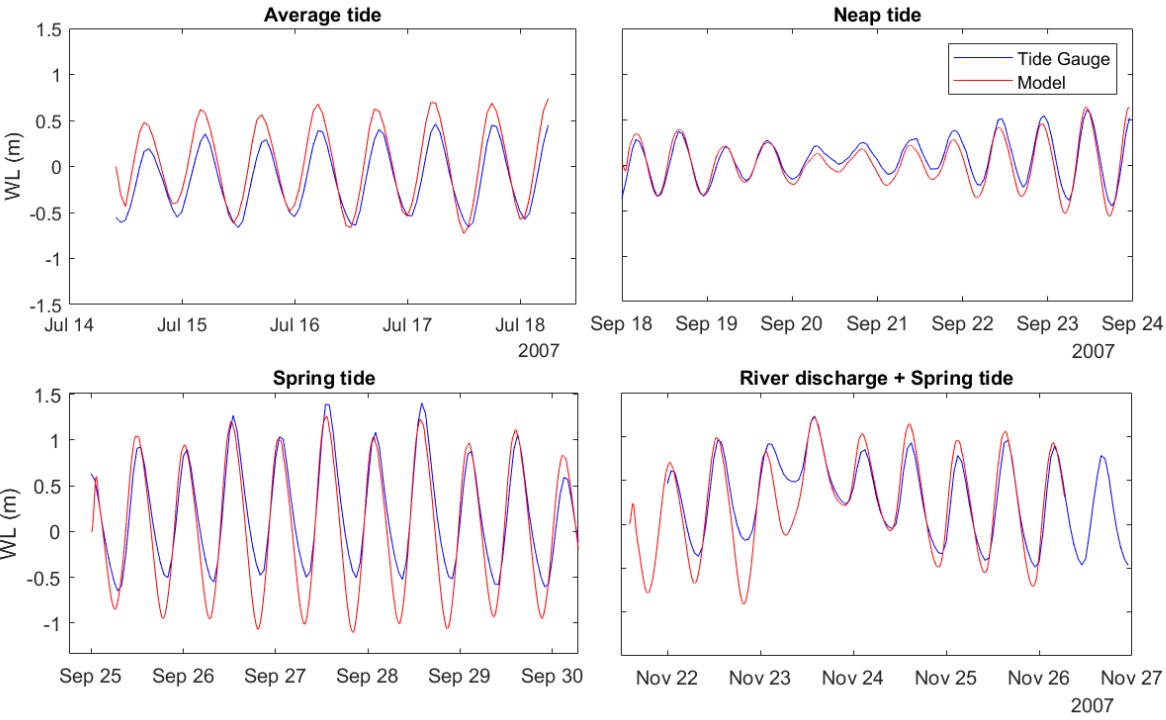

**Figure 3.** WLs of the model validation runs (red curve) at the tide gauge station, compared to observed WLs from the tide gauge (blue curve). Upper left panel shows WLs of the average tide event, upper right panel the neap tide event, the lower left panel the spring tide event, and the lower left panel the high river discharge event, coinciding with the spring high tide.

## 4.2 Flood sensitivity to varying driver combinations

To analyse the scenario results according to their flood characteristics in terms of magnitude and timing and to estimate the wave contribution, we initially compared the compound flood scenario $S_{TWQ}$ to scenarios in which one driver was excluded ($S_{TW}$, $S_{TQ}$). Then we compared the compound flood scenario $S_{TWQ}$ with the extreme wave compound flood scenario ($S_{TQWextr}$). WLs, flood extent, and maximum and mean flood depths of all compound scenarios are summarised in Table D1 of Appendix D. For demonstrative reasons we separated the model domain into three areas, termed "upper", "centre" and "lower" domain as shown in Fig. 5.

The results of the compound flood simulation ($S_{TWQ}$) with the simulation excluding river discharge ($S_{TW}$) showed large differences in all flood characteristics. The WLs of $S_{TWQ}$ were substantially higher throughout the entire estuary than the WLs produced by accounting only for oceanic drivers ($S_{TW}$, Fig. 4).






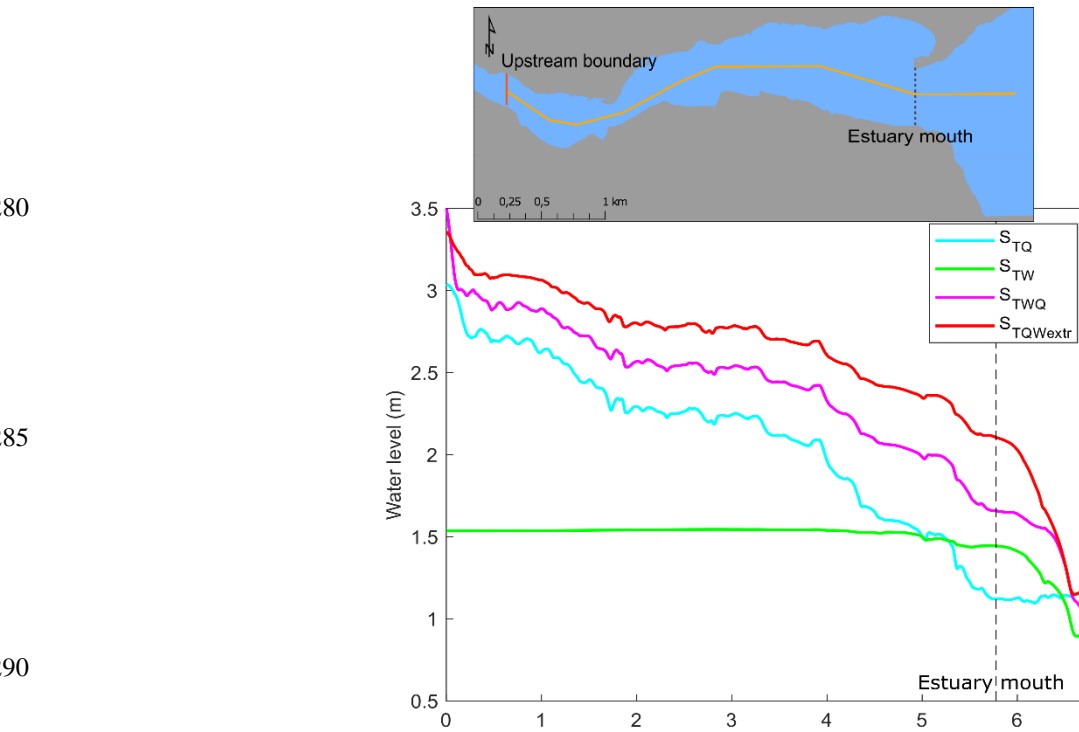



**Figure 4.** WL (m) with distance from the upstream boundary of all scenarios with the vertical dashed line demonstrating the location of the estuary mouth. The map shows the location of the transect (yellow line), as well as the location of the upstream open boundary (orange vertical line) and the estuary mouth (grey dashed line).

The WLs of $S_{TW}$ showed a continuous state around 1.54 m throughout the entire estuary, slightly decreasing towards the estuary mouth. As in $S_{TWQ}$, WLs were highest at the upstream open boundary and decreased substantially towards the estuary mouth,

the largest WL differences between both scenarios occurred at the upper domain with up to 1.5 m. Further towards the estuary mouth, differences reached a minimum of 15 cm. Outside the estuary, WLs of $S_{TWQ}$ were similarly higher, compared to the respective levels in $S_{TW}$.

Figure 5(a) presents the flood extent of $S_{TW}$ on top of the extent of $S_{TWQ}$, which showed a substantially larger extent. Further, both scenarios showed large spatial differences in flood extent patterns. $S_{TWQ}$ inundated an additional extent of 45%, compared

to the flood produced by the $S_{TW}$ scenario (Table D1). During the compound scenario, the flood covered a large low-lying area at the northern shore (about 5 km from the mouth), inundating up to 570 m further inland. However, in the scenario $S_{TW}$ where Q was excluded, only a narrow area got flooded, reaching at its widest part 250 m inland. On the southern bank (centre), the $S_{TWQ}$ flood reached 80 m further inland than $S_{TW}$. At the estuary mouth, both scenarios flooded about the same areas.

Figure 5(b) represents differences in flood depths. From the estuary mouth towards the estuary entrance, differences in flood

depths showed the same pattern as differences in WLs. At the sand barrier, flood depth differences reached up to 1 m.



Comparing WLs of the Q scenario in which waves were excluded ($S_{TQ}$) to the compound flood scenario $S_{TWQ}$ (Fig. 4), both WL curves showed the same pattern, with the WLs of $S_{TWQ}$ generally being higher than those simulated by $S_{TQ}$. The differences in WLs between both scenarios decreased from the area around the estuary mouth with maximum differences of 53 cm towards the centre of the study area. We found the smallest differences of ~20 cm close to the upstream edge of the model domain

where WLs were highest in both scenarios. We observed a similar pattern in flood depth differences (Fig. 5(e)), showing a maximum of 70 cm at the northern shore of the estuary entrance, decreasing towards the upstream boundary to ~20 cm. Figure 5(b) shows the overlaying flood extents of both scenarios, where both scenarios inundated mostly the same areas. The flood extent of $S_{TWQ}$ covered a 10% larger area than the flood, resulting from $S_{TQ}$ (see Table D1 for the flood size). Inside the estuary, the largest differences occurred in the populated area at the southern shore (centre).


**Figure 5.** Comparison of flood extents of compound and excluding driver scenarios (left panel, a), b) and c)) and differences in flood depths (right panel). Panel d) shows the flood depths of $S_{TWQ}$ - $S_{TW}$, e) shows $S_{TWQ}$ - $S_{TQ}$ and f) $S_{TQWextr}$ - $S_{TWQ}$.



As anticipated, both scenarios accounting for all three drivers during an extreme stage ($S_{TWQ}$ and $S_{TQWextr}$) showed the highest values in terms of inundation depth and extent. Comparing the compound flood scenario ($S_{TWQ}$), with the one including even higher extreme waves ($S_{TQWextr}$), we found large differences in the WLs, throughout the entire study area (Fig. 4).

Inside the estuary, $S_{TQWextr}$ produced continuously higher WLs, than $S_{TWQ}$, with increasing differences of up to 40 cm towards
the estuary entrance. Such differences are further encountered in the flood depth, showing the same magnitude in the entire lower area. Generally, the higher flood depths produced by $S_{TQWextr}$ reached towards the upstream open boundary, but the differences were decreasing (Fig. 5(f)). The flood extent was 12% larger, when considering large waves during compound flooding. Spatially, the larger flood plain in $S_{TQWextr}$ was mainly restricted to the southern shore of the central and lower model domain. In these areas, the $S_{TQWextr}$ extent expanded up to 40 m further inland than the extent of $S_{TWQ}$. At the northern shore,
the only noticeable area, which got flooded in $S_{TQWextr}$, but not in $S_{TWQ}$, was the sand barrier forming the estuary mouth. $S_{TQWextr}$ almost entirely flooded the sand dune, indicating that it is likely to be eroded during a flood (Fig. 5(c)).

To further estimate the effects of waves during compound flooding on the timing of the flood, different time steps of the flood WLs in scenarios $S_{TQWextr}$, $S_{TWQ}$ and $S_{TQ}$ are presented in Fig. 6. The left panel shows all three scenarios at the same time step (17 March 2007, 23:45), which was selected according to the onset of high WLs at the upstream open boundary in $S_{TQ}$. The
three scenarios at the same time step showed the highest WLs at the upper model domain, which then decreased towards the open sea. Generally, $S_{TQ}$ produced the lowest WLs (Fig. 6(a)), followed by $S_{TWQ}$ (Fig. 6(b)), and the largest WLs were produced in $S_{TQWextr}$ (Fig. 6(c)).

The figure also reveals differences in the areas in which the high WLs dominated at that time. While in $S_{TQ}$ WLs of up to 1.8 m were only shown in the upper area, the same magnitude of WLs reached until 4.2 km in $S_{TQW}$ and even crossed the estuary
mouth in $S_{TQWextr}$. Furthermore, the right panel of Fig. 6 shows at which time step two relatively large areas, marked by red boxes, were flooded in the three scenarios. As the model outputs were only saved every 15 minutes, it was not possible to extract the results at exactly the same flood stage, which is the reason for the slightly varying WLs of the scenarios. In $S_{TQWextr}$ the area got flooded earliest (18 March at 11:15am, Fig. 6(f)) and was followed 45 min later by $S_{TQW}$ (Fig. 6(e)). In $S_{TQ}$ however, the same area got flooded even 12 hours later, at the 19 March at midnight.



**Figure 6.** WLs of the scenarios $S_{TQ}$, $S_{TWQ}$ and $S_{TQWextr}$, extracted at the same time step (left panel) and extracted at the time step, where in each scenario marked areas (red box) were flooded for the first time during the entire simulation (right panel).


## 5. Discussion

### 5.1 Effects of interaction between drivers during compound flooding and the contribution of extreme waves

Model outputs show differences in the magnitude and spatial variation of flood characteristics between all scenarios. Spatial variations in flood characteristics of the different scenarios indicate locations where the interaction of waves, tides, and Q

during compound flooding have amplified flooding and where individual drivers contribute to the flood. Enhanced flood characteristics during compound flooding and spatial variations in the flood pattern caused by different driver combinations were previously discussed by Olbert et al. (2017), Kumbier et al. (2018) as well as by Bilskie and Hagen (2018). Yet, none of





the studies accounted for the additional influence of waves. In addition to this, none of them addressed the effects of the oceanic flood drivers on the timing of the flood, when co-occurring with Q.

The comparison of $S_{TWQ}$, with the riverine ($S_{TQ}$) and the wave scenario ($S_{TW}$), highlights that compound flooding increases flood extent and depth. In particular, the additional extent in the central study area, as well as the continuously higher WLs and water depths during compound flooding (Fig. 5(a), (b), (c), (d)) indicate an accumulation of water inside the estuary. The results further reveal where each driver has its highest influence. This information is relevant for understanding the flood dynamics due to driver interaction and the wave contribution. Regions only inundated in the compound flood scenario, but not

in $S_{TW}$, or $S_{TQ}$, were mostly located in the central zone of the study area. In $S_{TWQ}$ additional inundated areas in the upper sector were small (10%) when compared to $S_{TQ}$, but were large (45%) when compared to $S_{TW}$. These floods highlight the generally higher effect of river discharge in the more confined upper section during compound flooding. The influence of Q decreases towards the mouth area, as increased friction through the widening of the estuary at the central area and the large flood plain at the upper northern shore of the domain attenuate the flood wave (Cai et al., 2016). On the contrary, waves have clearly

shown to be the dominating factor at the estuary mouth area, resulting in substantially higher WLs (Fig. 4). These can be caused by wave set-up, as the steep bathymetry and shallow water depths outside the estuary cause waves to break before entering (Carter, 1983; Xu et al., 2020), increasing WLs inside the estuary (Olabarrieta et al., 2011; Zaki et al., 2015). Additionally, the funnelling effect due to the narrow estuary mouth may amplify wave set-up-induced WLs (Lyddon et al., 2018), contributing to the elevated WLs inside the estuary, and causing a relatively large flood extent at the sand barrier in

$S_{TW}$. The small differences in flood characteristics in the upper area ($S_{TWQ}$ vs. $S_{TQ}$), however, demonstrate a decreasing influence of waves from the entrance towards the upstream boundary of the model domain.

In $S_{TWQ}$ the increased WLs at the entrance and the larger flood extents at the sand barrier and in the central estuary indicate an interaction of drivers mostly in the lower area. Delpey et al. (2014) have shown that extreme waves can reduce the freshwater

outflow from the estuary mouth towards the open ocean, increasing the water volume inside the estuary, and thereby raise the WLs. Such a blocking of the riverine component through the oceanic component was also observed in Orton et al. (2020), although they only accounted for tides and excluded waves. Hence, the blocking of Q through waves may explain the larger flood characteristics in $S_{TWQ}$ at the central domain area, even approximating the upstream open model boundary. This shows a large contribution of waves on flood characteristics during compound flooding, which were not apparent when considering

the components individually. High outflowing Q can also dampen the wave and tidal propagation inside the estuary, causing increased WLs at the entrance (Sassi and Hoitink, 2013). This implies that during compound flooding, waves play a stronger role when coinciding with Q, by amplifying the flood impacts. When considering flooding individually however, the effects caused by waves were relatively low, as compared to effects caused by Q.

We further assessed the wave contribution by testing the sensitivity of compound flooding to more extreme wave conditions. Comparing results of the compound flood scenario $S_{TWQ}$ with results of the extreme wave compound flood ($S_{TQWextr}$) confirms





the expected larger flooding caused by more intense waves. This was valid for all flood characteristics throughout the entire study area. The effects of increased wave conditions were found to be greatest in the lower reaches. First, the considerably larger flood extent at the sand barrier can be explained by wave overtopping, and shows that extreme wave conditions coinciding with spring high tides may lead to eroding and a breaching of the barrier.

As explained above, wave set-up can raise WLs inside the estuary (Olabarrieta et al., 2011), which becomes more extreme with higher waves. An impact of waves on WL variabilities in South African estuaries has previously been shown by Schumann (2013) who states that waves together with the tidal influence can determine how far ocean water propagates upstream in an estuary. Therefore, increased wave conditions during compound flooding does not only have effects on the lower domain flood extent and depth. $S_{TQWextr}$ enhanced flood characteristics in the upper domain, as shown in all $S_{TQWextr}$ related figures (Fig. 4,

5(c), (f), and 6(c), (f)), confirm the fact that higher waves cause greater impacts further upstream when compared to $S_{TWQ}$.

Last, an interesting finding of the study is that compound events do not only affect flood characteristics in terms of magnitude. The timing of the flood also changed when all drivers coincide, and when stronger wave conditions are accounted for. The

increased volumes of water during the compound event and in $S_{TQWextr}$ resulted in flooding occurring earlier than when the drivers, waves, and Q, were not coinciding. Figure 6 shows that at the specific considered time step, the flood characteristics of $S_{TQWextr}$ were largest (also in the upper area), although at that time Q was still moderate. Therefore, even when the riverine component was still moderate, waves led to enhanced flooding. Considering the timing at which most of the in Fig. 6 marked flood plain was inundated in all three scenarios (see Fig. 6(d)-(f)) further highlights the large wave contribution during

compound flooding. In this case the flood plain was flooded earliest in $S_{TQWextr}$, followed by $S_{TWQ}$ about 45 minutes later. When not accounting for waves, however, the flood plain was inundated 12 hours later than in $S_{TWQ}$. These findings indicate that waves play a substantial role when coinciding with the fluvial component and spring tides, as they lead to larger flooding and an earlier onset of the inundation, even when Q was still moderate.

**5.2 Model performance, limitations, and outlook**

This analysis has shown the sensitivity of flood characteristics to compound flooding when compared to individual flood drivers. This was demonstrated by spatial variabilities in the flood extents, and by variabilities in the flood magnitude and timing. We must note however, that flood extent and depths could not be directly validated. Commonly used data types for flood impact validation are pictures, satellite imagery or high watermarks (Molinari et al., 2017). Yet, such data were not available for the study area. According to Basson et al. (2017), pictures and high watermarks of a fluvial flood, occurring in

2008 exist, however only at sites further upstream. This area was not considered in the model domain of this study, as detailed upstream bathymetry data could not be obtained. Nevertheless, model performance was validated at the tide gauge at Witsand near the mouth (Fig. 2). For all simulations including tides, the model demonstrated a correlation of 0.9. Flood peaks matched the observed peaks in almost all simulations (Fig. 3). The model overestimated tidal high-water peaks only during the average tide event. Tidal low water peaks, though, were generally underestimated.





Those differences can be explained by uncertainties inherent in the model input data, such as tides and bathymetry. Tides, serving as input data for model validation and all scenarios of this study were obtained from the global FES2014 tidal model (Carrère et al., 2015). Even though the model shows a rather high accuracy offshore, on shelves and on nearshore areas (Stammer et al., 2014; Ray et al., 2019), the local scale coastal processes caused by the local topography and the influence of the estuarine channel morphology (Wang et al., 2019; Godin and Martínez, 1994) are not considered in the data. Additionally,
the model open boundary was placed at a location several kilometers further nearshore of the point from which the tidal inputs were extracted. Processes modifying the tidal propagation between both locations were therefore not considered.

Additionally, the omitted storm surge, wind, and waves as model input during the validation runs can explain the large discrepancies occurring specifically in the Q validation run (Fig. 3, lower right panel), where tidal low water peaks preceding the actual event peak were strongly underestimated in the model. Thus, the higher observed WLs could have been produced
by wave set-up or less likely a storm surge (Zaki et al., 2015). Relative to other flood drivers, storm surge alone does not have a significant effect on coastal flooding along the South African south and west coast (Theron et al., 2014), but it still may effect WLs inside the estuary (Lyddon et al., 2018). Testing the effect of waves and surge on the model performance, however, would require observed wave time series and nearshore WL data which were not available for this study.

Storm surge has been considered in most regional or local flood assessments, specifically in those dealing with compound
flooding (Eilander et al., 2020; Olbert et al., 2017; Shen et al., 2019). Despite its low contribution at the location of Breede Estuary (Appendix C), storm surge may still contribute to compounding drivers to become an extreme event, even when neither of the drivers is extreme (Leonard et al., 2014). To estimate its contribution, storm surge should be considered in further simulations.

## 6. Conclusions

We assessed compound flooding from tides, Q, and waves at the permanently open Breede Estuary (South Africa) using a hydrodynamic model. For the assessment, we simulated scenarios accounting for the three flood drivers (i.e., tides, Q and waves) and scenarios omitting either waves or Q in order to analyse their contribution. We found that flood characteristics such as extent, water depth and timing are affected by the interaction between the drivers. As anticipated, the omission of waves caused major inundations to occur in the upper domain area, whereas the omission of Q produced comparably small
flooding. Thus, we have shown that when considered separately, the contribution of waves to flooding was small. When waves were combined with spring tides and Q however, they had a substantial effect on the spatial distribution and magnitude of the floods by impeding river flow to the sea. A notable impact of waves during compound flooding was their effect on the flood timing. Through backwater effects, waves induced the flood to occur earlier. This was further emphasised when increasing the wave intensity in the compound flood scenario. We therefore suggest that compound flooding induced by high Q, tides, and
waves should not only be considered in risk assessment studies in terms of magnitude, but also in terms of timing. The earlier onset of intense flooding needs to be accounted for when forecasting, planning, and managing flood hazards.





As we have shown in this study for Breede Estuary, compound flooding can exacerbate flooding and waves make a substantial contribution to flooding, when coinciding with extreme Q. Extreme waves co-occurring with spring tides and high precipitation have been documented by Guastella and Rossouw (2012), who additionally predicted a change in wave climate for the South

African south-west coast towards more frequent extreme wave conditions. Our results in combination with a changing wave climate further confirm the necessity of accounting for compound flooding and specifically waves in future local flood impact assessments in South Africa, particularly for other South African estuaries, which are highly populated, as Mngeni Estuary (Durban), Swartkops Estuary (Port Elizabeth), Nahoon Estuary (East London), Diep/Rietvlei (Cape Town) where it can lead to substantial infrastructure damage.

The achievement of data for complex modelling studies, as well as validating model results, in South Africa remains a major challenge, however.

**Appendix A: Data pre-processing**

For the hydrodynamic model we used the 5 m SUDEM elevation dataset (van Niekerk, 2016) merged with bathymetric data, which we manually digitised, based on a bathymetry of a study report on flood lines at the Breede Estuary (Basson et al.,

2017). As model friction parameters, we specified spatially varying manning values from land cover raster data, provided by the Department of Environmental Affairs (DEA, 2014), originally coming in a 30 m horizontal resolution. Manning roughness values for the different land cover classes were derived from a literature review, following Kaiser et al. (2011), Jung et al. (2011), Wamsley et al. (2009), Mourato et al. (2017) and Chow (1959).

As model boundary conditions we used historical tidal input data, which we obtained from the global tidal FES2014 model

(AVISO, n.d; Carrère et al., 2015). We extracted the data at a point closest to, but still located 24 km offshore from the open boundary. The time series covers a period of 34 years, from 1980-2014. We obtained hourly measured river discharge from the station H7H006 in Swellendam, which was the closest to the upstream boundary (54 km). The data were provided by the Department for Water and Sanitation of South Africa (DWS, n.d.b) and cover a period of 53 years, from 1966 until 2019. Hourly water level observations serving for the model calibration and validation were provided by the DWS from the tide

gauge station H14T007 (DWS, n.d.a), located in a small harbour of the town Witsand, inside the estuary mouth. The measurements cover 17 years (2002-2019). We derived wave data, significant wave height ($H_s$) and peak period ($T_p$) from two extreme value analysis (EVA) performed by Basson et al. (2017). They extracted from the ECMWF model simulated offshore wave data of 37 years (1979-2016), from a point close to the estuary, while still being located 30 km off the coast.

**Appendix B: Model set-up and model calibration**

Grid and topography of the model are based on the Cartesian coordinate system WGS84/UTM zone 34S. The model domain expands over an area of 19.2 km², covering the lower estuary and the area until 1.5 km offshore (Fig. 2). We used a time step



of 1.5 seconds for calibration, validation and scenario runs, as it was suggested by the Courant number. We changed the reflection parameter α, which determined the permeability of the open boundaries to 1000 for the ocean boundary and to 200 for the upstream boundary, as the model otherwise produced instabilities in preliminary runs (Deltares, 2014a).

Additionally, we considered several physical and numerical parameters for the model set-up. These were either kept at the default value, as suggested by Deltares (2014), or were changed and adjusted during model calibration runs. For the wave set-up we increased the grid cell size to 10 m for computational reasons. We used a JONSWAP (Joint North Sea Wave Project) spectrum with a peak enhancement factor of 1.75 and a wave direction spreading of 30° according to Basson et al. (2017).

For the model calibration we selected an event occurring from the 26 June until the 3 July 2003 due to the low and constant river discharge before, during and after the calibration event (max 3 m³s$^{-1}$). This is important, because the time lag of the river discharge between the measuring station in Swellendam and the upstream open boundary of the model domain is not considered. Waves were excluded during model calibration and validation.

For the model calibration we changed the physical parameters bottom roughness and horizontal eddy viscosity, as these can
affect the tidal amplitude and the speed of the tidal wave propagation into the estuary (Skinner et al., 2015; Garzon and Ferreira, 2016). Table B1 shows changed physical parameters and goodness of fit estimates, resulting from compared modelled and observed time series. The best fitting physical parameters resulting from the calibration were used to set-up the model for the validation and scenario runs, even though the improvements were small (see Table B1).

**Table B1.** Parameter changes during model calibration and final model set-up

| Model calibration | | | | |
|---|---|---|---|---|
| **Goodness of fit** | **Default** | **$n = 0.035$** | **$n = $ Land cover** | **Visc* = 4 m²/s** |
| RMSE | 0.22 m | 0.21 m | 0.21 m | 0.21 m |
| R² | 0.76 | 0.77 | 0.77 | 0.77 |
| r | 0.96 | 0.95 | 0.96 | 0.96 |
| **Final model set-up** | | | | |
| **Simulation** | **Resolution** | **$n$** | **Visc (m²/s)** | **Time step (s)** | **Alpha** |
| Calibration & Validation | 5 m | Land cover | 4 | 1.5 | 1000 |
| **Delft3D module** | **Resolution** | **$n$** | **Visc (m²/s)** | **Time step (s)** | **Alpha** |
| FLOW | 5 m | Land cover | 4 | 1.5 | 1000/200 |
| WAVE | 10 m | Land cover | 4 | 1.5 | 1000 |

## Appendix C: Surge contribution

To estimate storm surge height at Breede Estuary, we extracted the non-tidal residual (NTR) of the Witsand tide gauge time series. Then we performed a harmonic analysis on the water levels, using the Utide package of Codiga (2011) and subtracted the resulting tidal signal from the tide gauge data. The tidal signal plotted against the NTR is shown in Fig. C1. We calculated



the mean amplitude of the entire NTR time series (0.1 m) and the mean peak height of all NTR peaks, including outliers, being

0.54 m. Only several outliers exceed the average peak height of the NTR, reaching up to 1.7 m (Fig. C1). As the NTR still

contains the signal of river discharge from Breede River, we tested if peaks can be related to river discharge. Thus, we tested,

if NTR peaks occurred within 3 days after peaks of in Swellendam measured river discharge time series. In total 9 NTR peaks

were considered as being caused by high river discharge, of which 5 are highlighted in Fig. C2, for the period 2007 - 2010.

Additional to river discharge and non-linear interactions, the NTR at Witsand includes wave set-up. The contribution of wave

set-up to coastal water levels has already been shown by Dodet et al. (2019). As for our local study no measured nearshore

wave time series could be obtained neither for the study area, nor for close by locations, it is difficult to estimate the

contribution of wave set-up. In a widely applied formula wave set-up has been estimated to be 0.2x $H_s$ (e.g., Vousdoukas et

al., 2016). As according to Basson et al. (2017) and Guastella and Rossouw (2012) $H_s$ can exceed 10 m (100-yr return period)

at the area, Breede Estuary is located at, we can according to named wave set-up estimations assume that the component

contributes a substantial proportion to the NTR, underlining the assumption of a comparably small surge contribution.


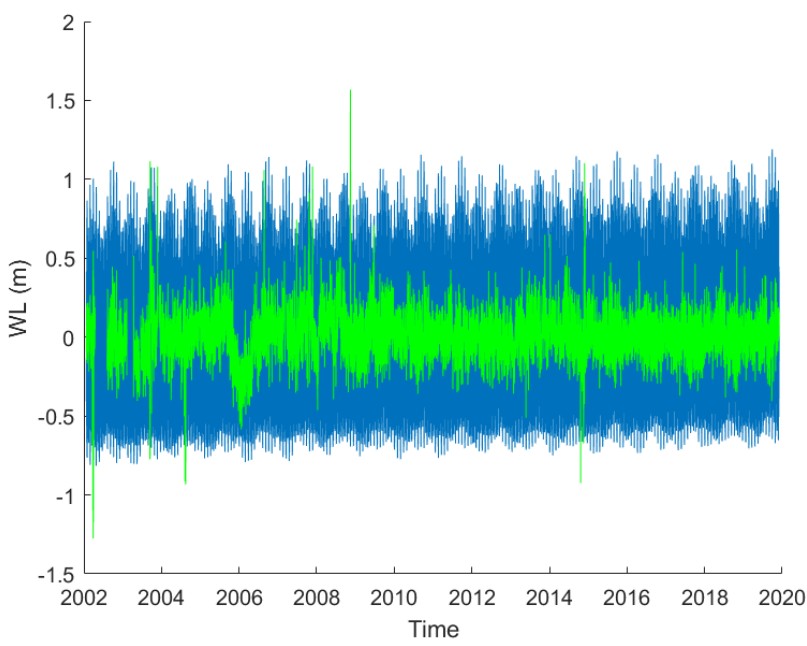


525                              **Figure C1.** NTR (green) plotted against tidal signal (blue) at Witsand




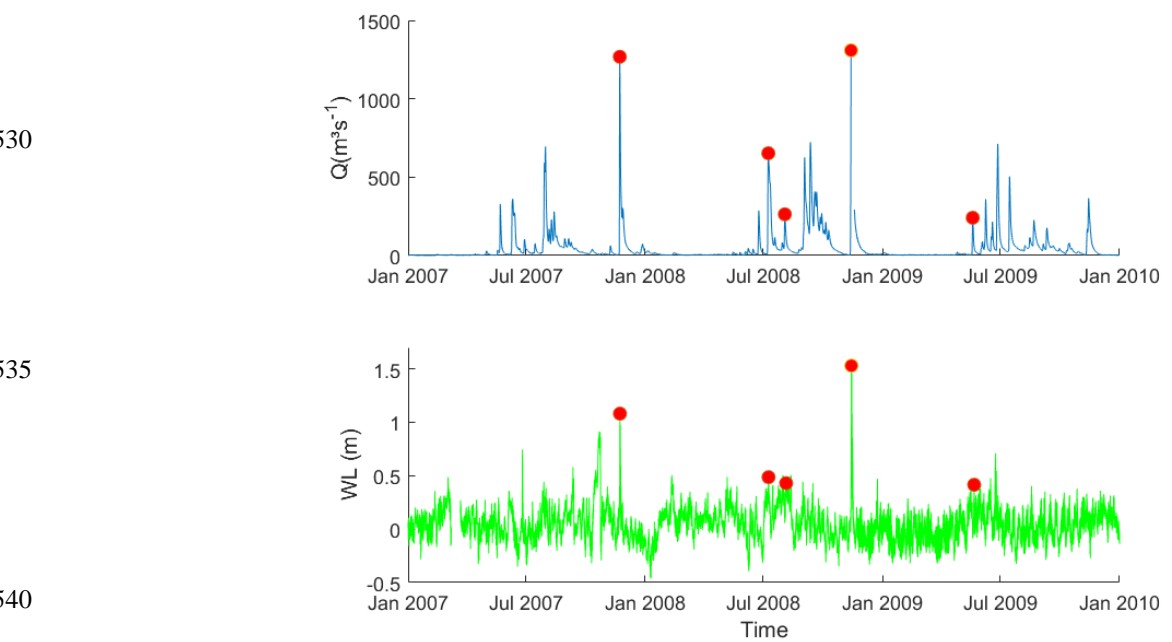

**Figure C2.** River discharge for the period 2007 - 2010 with peaks (red markers) occurring within 3 days before peaks of NTR (upper panel). NTR at Witsand with peaks (red markers) occurring within 3 days after peaks of river discharge (lower panel). The period 2007 - 2010 was chosen for representative reasons, as this was the period containing most peaks.





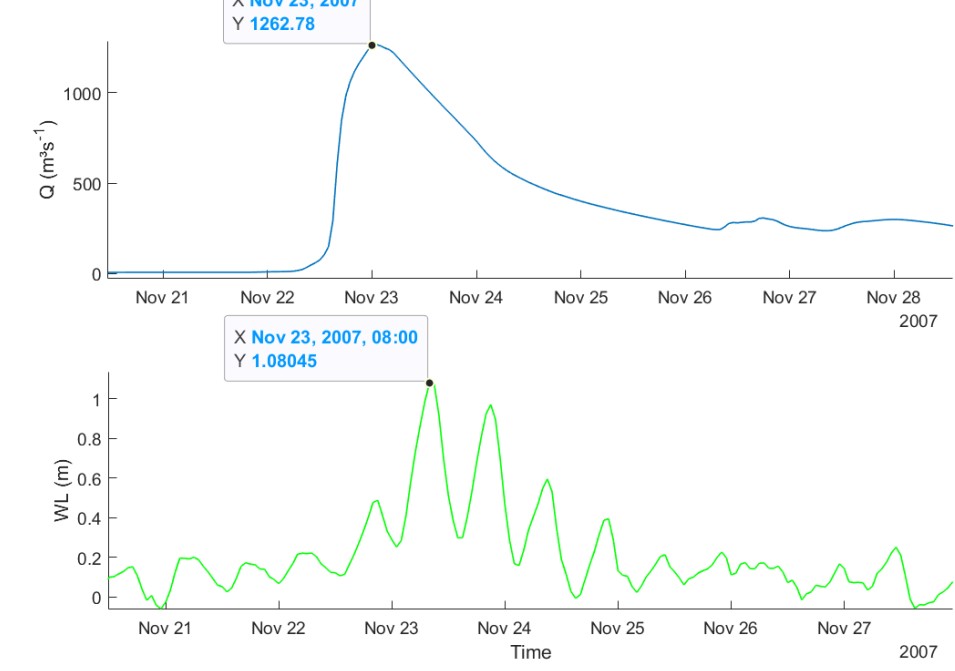

**Figure C3.** Validation event (22 - 25 November 2007) of river discharge at the flow gauge Swellendam (upper panel) and of the NTR at the tide gauge in Witsand (lower panel), between both peaks we estimated a time lag of 8 hours.



## Appendix D: Results

**Table D1.** Flood extents and maximum and mean flood depths of all scenarios

| Scenario | Flood extent (km²) | Mean. flood depth (m) | Max flood depth (m) |
|----------|--------------------|-----------------------|---------------------|
| $S_{TQ}$ | 0.66 | 1.06 | 3.61 |
| $S_{TW}$ | 0.46 | 0.71 | 3.52 |
| $S_{TWQ}$ | 0.73 | 1.28 | 4.08 |
| $S_{TQWextr}$ | 0.82 | 1.42 | 4.45 |

*Data availability.* All data used in this paper are properly cited and referred to in the reference list. The local tide gauge and flow gauge data can be obtained under request from the DWS, or can be downloaded from this webpage: https://www.dws.gov.za/Hydrology/Verified/hymain.aspx. FES2014 global tide model data are publicly available at https://www.aviso.altimetry.fr/es/data/products/auxiliary-products/global-tide-fes.html. The SUDEM can be purchased from Stellenbosh University for research purposes at https://geosmart.space/products/sudem.html. Bathymetric data and model output can be provided from the first author on request.

*Author contribution.* SK defined the research problem, collected the data, performed all the analyses, prepared the figures and wrote the first draft of the manuscript. SSA provided advice on the methodology and SSA and AV supervised the work of SK. LN and MLV helped with finding the study area and LN and MLV provided relevant literature and links to hosted datasets. SSA, AV, MLV and LN reviewed the manuscript.

*Competing interests.* The authors declare that they have no conflict of interest.

*Acknowledgements.* This research has been supported by the German Federal Ministry of Education and Research (grant SPACES-CASISAC (03F0796B)). Lara van Niekerk and Melanie Lück-Vogel were funded through the Coastal Systems CSIR Parliamentary Grant (PG) from the Department of Science and Innovation (DSI).

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
