# Peer review of "Investigating the interaction of waves and river discharge during compound flooding at Breede Estuary, South Africa"

_Natural Hazards and Earth System Sciences, 2021_

## Referee Comment (RC2)

**Review of "Investigating the interaction of waves and river discharge during compound flooding at Breede Estuary, South Africa" by Kupfer et al.**

The paper investigates the interaction between tide, discharge and waves with a focus on waves to analyse compound flooding in the Breede Estuary. The study is based on a scenario analysis using the numerical Delft3D model. The paper highlights the importance of waves for compound flooding for South African open estuaries based on the scenarios used. The paper is well written and explains the results in a concise way. However, the study leans heavily on the selection of scenarios which comes with some limitations which should be better addressed in the discussion and some choices that should be better justified in the methods section. I therefore recommend the publication after major revisions.

**General comments**

The selection is rather limited and conservative based on the combination of 100-year return values of Q and waves without any analysis of the dependence between Q and waves. While I appreciate that adding a statistical analysis would change the focus of the paper and modelling many more combinations of flood drivers with different return periods will dramatically increase the computational effort, it would provide a more comprehensive analysis of the importance of waves for compound flood risk. At least some discussion on choice of scenarios on the results should be included.

It is also unclear what the relative timing between the Q, waves and tides peaks is and how this is selected and what the effect of the time lag on the results is. If no observed data is available for a time lag analysis a sensitivity analysis would strongly improve the robustness of the findings.

**Specific comments**

L185: A table for the validation events would be helpful to a reader.

L226: It is unclear how the return value for Q is translated into an event hydrograph which is required to force the model.

L227: It is unclear why the maximum Q peak value was corrected. Also provide more details about how the 100-year event was derived from the time series.

L229: Can you justify the choice of a 3 m3/s Q event for the $S_{TW}$ scenario?

L244: It is not clear how the two columns in Table 3 relate to the four events shown in Figure 3. It would be useful to include all events separately in table 2 or use the same naming or numbering for the events in both. A table of all validation events as mentioned earlier would also help.

L296: It is not clear what you mean with "similarly higher". In Figure 4 it seems to me that the difference between $S_{TW}$ and $S_{TWQ}$ decreases outside the estuary.

L335: A simple time series plot of WLs at one or more locations would be easier and more appropriate and informative instead of the right panel of Fig 6.

L377: Should "flooding" not be "flood drivers" in this sentence?

---

## Author Response (AR1)

**Responses to reviewers:**

We would like to thank both reviewers for their detailed reviews of our manuscript and for the relevant points that they have raised.

In the context of our response, we have conducted some further analysis and addressed all the points that were raised by the reviewers. In response to the suggestions, the main changes in the revised version of the manuscript are:

1) We have included a literature review on the wave contribution to WLs.
2) We have added a table summarising the events used for validation.
3) We have added goodness-of-fit estimates for the model validation and modified table 3.
4) We have updated figure 3.
5) We have elaborated in more detail on the scenario choice and dependence analysis in the methods and discussion.
6) We have conducted additional scenarios.
7) We have updated the graphical representation of figure 5 and figure 6.

We believe that the manuscript has benefited substantially from the revisions and hope that the reviewers agree with the changes that we have implemented.

Please find below the answers to the individual comments and suggestions (in blue bold font) of both reviewers. The answers to the comments are placed below the comments (in black font) and changes made in the manuscript are included in italic black font.

**Reviewer 1:**

**As the main focus/novelty is the contribution of waves to WLs, more overview of literature and their findings on that matter would be of benefit to a reader.**

We agree that a literature overview on the contribution of waves to WLs further highlights the focus of the manuscript and provides relevant background information to the reader. We have therefore added a paragraph from line 52 in the introduction as follows:

*"Waves can raise WLs at the coast in terms of wave set-up, which is described in detail by Dodet et al. (2019). Tanaka et al. (2009) have shown that in a shallow and narrow estuary entrance, wave set-up can be up to 14% of the offshore wave height. For South Africa, Marcos et al. (2019) have shown a dependence of extreme water levels and waves, and according to Melet et al. (2018) and Theron et al. (2010) waves constitute the most important components of coastal flooding for the country. Large destructive swells are generated by cold fronts, cut-off lows, and cyclones (Guastella and Rossouw, 2012). These low-pressure systems cause additional heavy rainfalls, leading to immense fluvial flash floods (Pyle and Jacobs, 2016; Molekwa, 2013). Thus, a dependency between both drivers is likely. However, no published regional to local compound flood probability analyses exist for South Africa…."*

**Also, a discussion and comparative analysis with the authors' findings would be of use in the discussion section.**

We understand and agree that a more extensive discussion and a comparative analysis with other findings on the wave contribution to water levels during compound flooding would provide further understanding to the reader. Unfortunately, we find a results comparison difficult as there is no published literature quantifying compound flooding from waves, river discharge and tides in an estuary.

**The performance of hydrodynamic model is arguable as some deficiencies in model accuracy in terms of water elevations exist (velocity magnitudes are not validated). The validation results show some significant discrepancies between the model and observations, this is further confirmed by RMSEs, which for tide+Q WL is 0.62. While this number is very high in the flood context, it seems from fig 3 that these discrepancies are primarily generated on low water peaks of spring conditions while high water peaks (which are of relevance in this study) are quite comparable. The separate analyses of RMSE for neap, spring tides and low and high water would be useful in the process of validation.**

Thank you for this comment. Following your suggestion and the suggestion of the second reviewer, we have estimated the goodness-of-fit for WL peaks of each individual event used for model validation. The results of the additional statistical analysis are presented in the following table. We moved this table to Appendix C, as we included the additionally calculated goodness-of-fit estimates for the peak values of each individual tidal event in Figure 3.

**Table C1.** Goodness-of-fit estimates of model validation runs compared to observations. Columns 2-4 show goodness-of-fit estimates for each tidal event of flood peaks only. Column 5 shows the goodness-of-fit for tide-only conditions (entire time series) and column 6 for tides including high river discharge (entire time series).

| Goodness-of-fit | Average | Neap | Spring | Spring + high Q | Spring-, neap-, average | Spring-, neap-, average & Spring + high Q |
|---|---|---|---|---|---|---|
| RMSE | 0.25 | 0.07 | 0.14 | 0.11 | 0.21 m | 0.23 m |
| $R^2$ | 0.52 | 0.79 | 0.78 | 0.69 | 0.8 | 0.94 |
| r | 0.96 | 0.91 | 0.85 | 0.81 | 0.9 | 0.91 |

[Figure]

**Figure 3.** WLs of the model validation runs (red curve) at the tide gauge station, compared to observed WLs from the tide gauge (blue curve). (a) shows WLs of the average tide event, (b) the neap tide event, (c) the spring tide event, and (d) the high river discharge event, coinciding with the spring high tide. All panels include goodness-of-fit estimates for peak values of each event ($RMSE_{peaks}$, $r_{peaks}$).

We hope that this analysis of RMSE for peak values brings more clarity regarding the source of the error and highlights model performance at peak values.

Moreover, we would like to particularly thank the reviewer for this comment as, by recalculating the statistical measures, following the comment below, we detected an error in the previous goodness-of-fit estimates for the "Spring-, neap-, average tide & Spring + high Q" analysis. We included the

corrected values in the table, which leads to large improvements in the model results. We also corrected the RMSE of all peak values, being now 0.15 cm, which we corrected in the manuscript in line 264.

**The discrepancies could be due to the number of factors including spatial resolution of FES2014 tidal constituents and a number of constituents being simulated, as well as not including the non-tidal signals. Nesting or downscaling the boundary conditions from the regional model would be more appropriate, however I understand that due to the computational effort and the timescales required here this may not be possible.**

Thank you for raising this interesting point. We must confirm that a nesting approach would require large computational effort and is therefore too challenging to realize within this short time period. However, we agree to add this point to the discussion. We have therefore added a sentence to the discussion in line 427:

"*One way to overcome this limitation would be to downscale the tides from the model towards the location of the open boundary, which, however, is beyond the scope of this study.*"

Further, errors may arise from the digitized bathymetry, as Basson et al. (2017), who generated the bathymetry in 2017, could not provide detailed information. Moreover, the inner estuarine area is a highly dynamic area containing many sandbanks. These may have been in different positions in 2007 (the year we extracted the validation events) compared to 2017, affecting the tidal inflow and constituents. Thus, we added a paragraph to the discussion, elaborating on this limitation in line 429:

"*Moreover, permanently opened estuaries are highly dynamic areas due to a constant influence of sediment deposition by river inflow and sediment removal due to floods (Moore et al., 2009; Whitfield et al., 2012). The sand bars and sand banks at the timing of the validation runs (covering events in 2007) were therefore likely in a different position than at the time when the input bathymetry was generated (Basson et al., 2017). This can have a high impact on water levels at the location of the tide gauge (Wang et al., 2019).*"

**Setting the north and south boundary as the no-flow wall boundary may play a role too.**

We would like to point out that our validation shows that the model is generally underestimating ebb tidal levels. Opening of the north and south boundary would lead to more water flowing out of the model domain. Therefore, we do not expect that open north and south downstream boundaries would improve model performance.

**Finally, one-hour resolution of river discharge timeseries at upstream boundary could also contribute to the error.**

We agree that the magnitude of river discharge can change on a higher frequency than hourly; however, no higher resolved data could be obtained.

**Also, it is not clear whether a 3D version of the model is used; the 3D-mode would be a better choice as the baroclinic conditions could be important in the region due to seasonal development of stratification.**

We used the depth-averaged mode the model validation, as well as scenario runs for computational reasons. The depth-averaged mode has been successfully applied in other flood modelling studies in estuaries, using Delft3D, such as Kumbier et al. (2018) as well as other hydrodynamic models used by

Skinner et al. (2015). In the method section, in which we describe the model setup we point out in line 158f:

*"We performed simulations […] in Delft3D-FLOW […] in a depth averaged mode for the model validation, as well as scenario runs. The 2D mode has been successfully applied in numerous hydrodynamic flood modelling studies (Kumbier et al., 2018; Skinner et al. 2015; Olbert et al. 2017)."*

**In Table 3, I would suggest to present statistical analysis separately for spring, neap and average (and perhaps for close to peak values only) to better understand the source of an error.**

We have referred to this comment in the previous response, showing table C1 and figure 3.

**In the results section, the four selected scenarios represent only a combination of the most extreme (100-year RP) conditions. As such, this is a very conservative approach and does not explore all flood conditions and their probabilities of occurrence. In my opinion a combination of varied Qs and oceanic conditions such as the 'marginal', 'and', 'or' scenarios would provide better understanding of interactions and perhaps their non-linear effects.**

We agree that accounting for extreme scenarios only does not explore a wide range of flood conditions, which would be a limitation when performing a risk assessment. However, we have decided on the extreme 'and', 'or' and 'combined' scenarios, for the following reasons: First, we follow recommendations of previous flood assessment studies for South Africa. Most have previously used 50-year return periods, which is now shifting towards using 100-year return periods (Theron and Rossouw 2008; Basson et al., 2017). For computational reasons we did not explore a wider range of scenarios, as lower wave return periods (20- and 50-year) do not significantly differ from the 100-year return level and higher return levels were not available. As our results clearly show that the estuary is Q-dominated, we did not consider higher return periods for river discharge. We added a sentence in section 3.3, line 246 to explain our choice of scenarios:

*"Due to computational constraints and data limitations, we have employed the 100-year return period for waves and Q, as this was also recommended by previous flood assessment studies for South Africa (e.g. Theron and Rossouw, 2008; Basson et al., 2017)."*

As we have shown that Breede Estuary is, even during compound flooding, a Q-dominated estuary, we developed an additional scenario with extreme wave (100-year, all directions) and moderate (20-year) Q conditions. However, the results did not significantly differ from the 100-year RP compound flood scenario, which is why we decided not to include the additional scenario in the manuscript.

We elaborate on the scenario decision in the discussion from line 445. Please find the additional paragraph in the comment below.

**The readers would benefit from dependence analysis between Q and WLs. In recent years the use of Kendalls tau and copulas to construct a bivariate models have been widely used in this type of analysis. I appreciate however, that such statistical analyses would require a large effort and lead to a substantially different paper. I would suggest to discuss the interactions and dependencies in context of future work.**

We strongly agree that dependency analysis between waves and Q would give an overview on the probability of co-occurring extreme conditions. However, we agree with the reviewer that focussing on a statistical dependency analysis would change the focus of the paper. We have indeed performed a preliminary dependence analysis between Q and waves by estimating the Kendall's Tau coefficient (see Ward et al., 2018; Couasnon et al., 2019). However, we decided not to present these results,

since we used data from two different sources, as wave observations were not available. To assess dependence however, data used should come from the same source (Marcos et al., 2019). We used observed Q and modelled offshore waves from the CSIRO hindcast, produced from NCEP-NCAR reanalysis data (Cox and Swail, 2001). Moreover, we did not include a dependence analysis in the manuscript, since independence or dependence does not affect our results, showing driver interaction. It would only provide an overview on the likelihood of occurrence of our chosen results, which is relevant for a risk assessment. However, we do agree that discussing dependencies in the context of future work would be useful. We therefore added a paragraph to the discussion in line 445:

*"Our analysis presents driver interactions during extreme (100-year) conditions, without showing joint probabilities of waves and Q. This information becomes relevant when assessing risk from compound flooding, which is beyond the scope of this study and should be considered in future work. For such a risk assessment, a wider range of return periods should be explored."*

**Figure 5, right panel. The colour scale could be refined to better reflect the differences. The interval of 0.3 is substantial in the flood context. The negative values in the range are not used.**

We thank you for this advice. We removed the negative values from the legend and decreased the interval to 0.1, as shown in the figure below:

[Figure]

**Figure 5.** Comparison of flood extents of compound and excluding driver scenarios (left panel, a), b) and c)) and differences in flood depths (right panel). Panel d) shows the flood depths of $S_{TWQ}$ - $S_{TW}$, e) shows $S_{TWQ}$ - $S_{TQ}$ and f) $S_{TQWextr}$ - $S_{TWQ}$.

**Moreover, there is a number of typos (line 37- semicolon etc) which should be removed.**

We have gone through the manuscript and have corrected all the errors we identified.

**Reviewer 2:**

**The selection is rather limited and conservative based on the combination of 100-year return values of Q and waves without any analysis of the dependence between Q and waves. While I appreciate that adding a statistical analysis would change the focus of the paper and modelling many more combinations with different return periods will dramatically increase computational effort, it would provide a more comprehensive analysis of the importance of waves for compound flood risk. At least some discussion on choice of scenarios on the results should be included.**

Thank you for this remark. We have indeed performed a preliminary dependence analysis between Q and waves by estimating the Kendall's Tau coefficient (see Ward et al., 2018; Couasnon et al., 2019). However, we decided not to present these results, since we used data from two different sources, as wave observations were not available. To assess dependence however, data used should come from the same source (Marcos et al. 2019). We used observed Q and modelled offshore waves from the CSIRO hindcast, produced from NCEP-NCAR reanalysis data (Cox and Swail, 2001). Moreover, we did not include a dependence analysis in the manuscript, since independence or dependence does not affect our results, showing driver interaction. It would only provide an overview on the likelihood of occurrence of our chosen results, which is relevant for a risk assessment. We decided on the extreme scenarios following previous flood assessment studies for South Africa. Most have previously used 50-year return periods, which is now shifting towards using 100-year return periods (Theron and Rossouw 2008; Basson et al., 2017). For computational reasons we did not explore a wider range of scenarios, as lower wave return periods (20- and 50-year) do not significantly differ from the 100-year return level and higher return levels were not available. As we have shown that Breede Estuary is a Q-dominated estuary, even during compound flooding, we developed an additional scenario with extreme wave (100-year, all directions) and moderate (20-year) Q conditions. However, the results did not significantly differ from the 100-year RP compound flood scenario, which is why we decided not to include the additional scenario in the manuscript.

As our results clearly show that the estuary is Q-dominated, we did not consider higher return periods for river discharge. We explain our scenario choice in section 3.3, line 246:

"*Due to computational constraints and data limitations, we decided on the 100-year return period for waves and Q, as this was also recommended by previous flood assessment studies for South Africa (e.g. Theron and Rossouw, 2008; Basson et al., 2017).* "

We have added a paragraph to line 452, further elaborating on this:

"*Our analysis presents driver interactions during extreme (100-year) conditions, without showing joint probabilities of waves and Q, but we have not considered dependence and joint probabilities of waves and Q. This information becomes relevant when assessing risk from compound flooding, which is beyond the scope of this study and should be considered in future work. For such a risk assessment, a wider range of return periods should be explored.*"

**It is also unclear what the relative timing between the Q, waves and tides peaks is and how this is selected and what the effect of the time lag on the results is. If no observed data is available for a time lag analysis a sensitivity analysis would strongly improve the robustness of the findings.**

We agree with the reviewer that the timing between Q, waves and tidal peaks is not clearly explained in the manuscript. For all scenarios, peaks of tides and Q (when considered) occur at the same time, while waves are constantly at peak conditions (due to constant wave conditions). We agree that conducting simulations with different peak timings (sensitivity analysis) would clearly increase the relevance of our results, but would also require immense additional computational effort. Since we are interested in maximum flood heights, we decided to simulate co-occurring peak values only. To clarify the timing between Q, waves and tides, we added a sentence in section 3.3, line 228:

*"All scenarios assume that the peaks of the drivers occur at the same time."*

**L185: A table for the validation events would be helpful to a reader.**

Thank you for the suggestion. We included a table in section 3.2, summarizing all events used for validation and their names, used in figure 3. We adjusted the text in section 3.2, line 187 accordingly.

*"To account for the full tidal range, these simulations include a spring-, average-, and a neap tide event (see Table 3 for the event names and dates of occurrence)."*

We removed lines 188f.

**Table 3.** Validation events and dates of occurrence.

| Event Name | Average | Neap | Spring | Spring + high Q |
|---|---|---|---|---|
| Date | 14-19/07/2007 | 18-23/09/2007 | 27/09-01/10/2007 | 22-25/11/2007 |

**L226: It is unclear how the return value for Q is translated into an event hydrograph, which is required to force the model.**

We agree with this comment. We added a sentence to line 233, in section 3.3, to elaborate on how we came up with the hydrograph:

*"We developed the Q-hydrograph to force the upstream open boundary by normalizing the hydrograph of the highest Q event, for which the full hydrograph was available. We then multiplied the normalized hydrograph with the 100-year peak value."*

**L227: It is unclear why the maximum Q peak value was corrected. Also, provide more details about how the 100-year event was derived from the time series.**

Referring to the correction of the maximum Q peak value, we added a sentence to line 229 in the manuscript:

*"The value was corrected, as for this event the flow gauging station stopped measuring before the peak was reached."*

Basson et al. (2017) explain in detail how they came up with the 100-year return level of the Q time series. We refer to their report in line 231 and hope that this point is now clearer.

**L229: Can you justify the choice of a 3 m³/s Q event for the $S_{TW}$ scenario?**

Thank you for raising this point. First, we must correct the value, as the selected maximum Q of the boundary conditions for the $S_{TW}$ scenario was not 3 m³/s but 1.2 m³/s. We apologize for this mistake. Second, we agree that adding an explanation on this choice helps to understand the scenarios. We had to keep the upstream open boundary open, as a closed boundary would have led to water accumulation inside the estuary. We therefore chose the lowest measured Q from the time series, to keep the scenarios realistic. We do not expect such low Q to have an effect on the scenario results. We added a sentence in line 234 to explain the choice of Q for the $S_{TW}$ scenario:

*"For the $S_{TW}$, so the no-Q scenario, we kept the upstream boundary open, so that incoming flood water does not accumulate there. Thus, we chose the lowest measured Q event from the time series to force the upstream open boundary, where Q does not exceed 1.2 m³/s."*

**L244: It is not clear how the two columns in Table 3 relate to the four events shown in Figure 3. It would be useful to include all events separately in table 2 or use the same naming or numbering for the events on both. A table of all validation evens as mentioned earlier would also help.**

Thank you for this suggestion. We adjusted the event names in Figure 3 accordingly to relate to the newly created Table 3. We additionally added goodness-of-fit estimates to the figure and moved Table 3 to Appendix C. Please see our changes in the figure below:

[Figure]

**Figure 3.** WLs of the model validation runs (red curve) at the tide gauge station, compared to observed WLs from the tide gauge (blue curve). (a) shows WLs of the average tide event, (b) the neap tide event, (c) the spring tide event, and (d) the high river discharge event, coinciding with the spring high tide. All panels include goodness-of-fit estimates for peak values of each event ($RMSE_{peaks}$, $r_{peaks}$).

**L296: It is not clear what you mean with "similarly higher". In Figure 4 it seems to me that the difference between STW and STWQ decreases outside the estuary.**

We agree that the difference between $S_{TW}$ and $S_{TWQ}$ decreases outside the estuary. Since we are not focussing on the outside area, we modified the sentence in line 299f as follows:

*"Further towards the estuary mouth differences reached a minimum of 15 cm, which decreased towards the outer area."*

**L335: A simple time series plot of WLs at one or more locations would be easier and more appropriate and informative instead of the right panel of Fig. 6.**

Thank you for this remark. We agree that a time series of the area highlighted at the right panel in Figure 6 is more intuitive and informative. We therefore replaced all figures of the right panel with time series of all three scenarios, at three different locations. These points are included in each map of the left panel, as shown below. We hope that this modification considerably improves the figure and provides a better insight on the flood timing. We adjusted the text in the manuscript from line 338 accordingly.

[Figure]

**Figure 6.** WL maps of the scenarios S$_{TQ}$, S$_{TWQ}$ and S$_{TQWextr}$ at the same time step (left panel) and time series of all three scenarios, time series of all three scenarios, showing the timing of the onset of the flood, extracted from the point, marked by the blue star.

**L377: Should "flooding" not be "flood drivers" in this sentence?**

We agree with the reviewer, this was a mistake. We corrected it accordingly.

**Additional references:**

Cox, A. T., and Swail, V. R.: A global wave hindcast over the period 1958–1997: validation and climate assessment. *Journal of Geophysical Research*, **106**(C2), 2313–2329, https://doi.org/10.1029/2001JC000301, 2001

---

## Author Response (AR2)

**Response to the reviewer:**

We would like to thank the reviewer for the additional comment and for the recommendation of publishing after minor revision.

Please find below our reply (in black) to the reviewer's comment (in blue):

*In my opinion the relative timing between the Q, T and W peaks is important in this context and I appreciate the fact that a full sensitivity analysis would be a substantial additional computational effort. However, I would suggest to briefly discuss the implications of your choice (all peaks at the same time) in section 5.1. You could refer to e.g. the recent paper by Harrison et al (2021) for more context on the matter.*

Thank you for the suggestion. We have added a short paragraph (line 438) on the relative timing to the discussion (section 5.2) and refer to the suggested paper. The paragraph reads:

*"Further, we assumed that maximum flooding occurs when all drivers peak simultaneously, and did not account for differences in the relative timing of all driver peaks. However, Harrison et al. (2021) conducted such a sensitivity analysis and found that in an estuary comparable, in size, to Breede Estuary this effect was negligible. However, as estuaries can also differ in aspects other than size (e.g. morphological characteristics), assessing the effect of the timing of the driver peaks could provide further insights on the flood mechanisms of compound flooding. This information, together with information on joint probabilities becomes relevant when assessing risk from compound flooding, which is beyond the scope of this study and should be considered in future work. For such a risk assessment a wider range of return periods should also be explored."*